# Children's experiences of play in digital spaces: A scoping review

**Fiona M. Loudoun**[1]*, **Bryan Boyle**[1], **Maria Larsson-Lund**[2]

**1** Department of Occupational Science and Occupational Therapy, University College Cork, Cork, Ireland,
**2** Division of Health, Medicine and Rehabilitation, Department of Health, Education, and Technology, Luleå
University of Technology, Luleå, Sweden

she These authors contributed equally to this work.
* floudoun@ucc.ie

## Abstract

The United Nations Convention on the Rights of the Child has substantiated play for play's sake, thus focusing on the doing or being of play rather than any potentially desired outcomes. Examining this type of play from the perspective of the child acknowledges children as meaning-makers. A scoping review was conducted to expose and map the extent of the evidence available in the emerging and diverse field of children's experiences of play in digital spaces. Specifically, the literature was examined with regards to relevance to children's everyday lives, the personal and ecological relevance, and the methods used. A systematic search of the literature over the past fifteen years found thirty-one articles appropriate for inclusion. The analysis of the literature revealed that the articles formed four categories of how play in digital spaces was approached: 'Videogames, behaviours, and societal norms', 'Videogames for its own sake', 'Videogames for learning', and 'Active Videogames for health promotion'. This scoping review has identified a lack of articles focusing on children's experiences of play in a digital space, and these perspectives are essential for parents, professionals, game designers, and policymakers alike to contribute to an enhanced understanding of the role of play in digital spaces.

## Introduction

Play and the rights of children and young people has been resolutely established through the United Nations Convention on the Rights of the Child [1]. Specifically, Article 31 outlines that every child has the *right* to participate in play or leisure activities that are appropriate for the age of the child with Article 12 highlighting the *right* of the child's voice to be heard. Globally, the UNCRC has firmly re-positioned children in society, acknowledging them not only as the holder of rights but also able to share and make meaning from their unique perspective of the world [2–4]. Play is widely recognised to benefit children's emotional, social, physical, and cognitive development [5,6] and is valued from educational, therapeutic, and developmental perspectives. However, the *doing* or *being* of play; ultimately play for play's sake focuses on the "process rather than the product"[7]. Participation in play is accepted as a child's primary activity and is characterised as being initiated, organised, and controlled by the child

**Competing interests:** The authors have declared
that no competing interests exist

themselves [8,9]. Based on Bronfenbrenner's ecology of human development, the validity of this free play needs to be examined within the context of which it is happening and where children are consistently expected to be [10–12]. This, therefore, demonstrates the value of ecological relevance to children's daily life. Research exploring the *doing* of play rather than as a mean for another purpose is emphasised to prioritise children's perspectives of their own self-chosen and directed play. Although, there has been an increase in interest in research examining the nature of autotelic play in diverse backgrounds and environments such as playgrounds [13], urban spaces [14] and in rural settings such as forests [15], there remains a context that has been largely ignored to date.

The rapid acceleration in the development and accessibility of technology and digital devices throughout the 21st century has shifted the opportunities, nature, and the spaces in which children can play. Children are engaging in play from simple puzzle games to more complex strategy games using a plethora of devices such as smart phones, tablets, laptops, and game consoles. These devices are providing new contexts for children to engage in play in digital spaces. Children can now enter a shared virtual space in which they can engage in social and collaborative play with friends whilst being remotely located. Despite the increase in opportunities available, a polarised and often negative discourse continues to exist regarding the use of digital devices in children's play. This remains largely from the perspective of adults and dominated by the negative consequences of increased screen time and sedentary behaviours [16]. Thus, an enhanced understanding of children's perspectives of their play in digital spaces is essential to distinguish this type of play. A recent report 'Playful by Design: A Vision of Free Play in a Digital World' from The Digital Future Commission identified 12 qualities of free play within a digital space valued by children: intrinsically motivated, voluntary; open-ended; imaginative; stimulating; emotionally resonant; social; diverse; risk-taking; safety; sense of achievement; and immersive [17]. This report emphasises the importance of recognising free play in digital spaces from the child's perspective and a shift to presenting the emerging knowledge to stakeholders.

By conceptualising children as meaning makers, the inclusion of their unique perspective of play throughout the literature is recognised. It has been identified that children do not distinguish between types of play as adults do [9,18,19], therefore further research *with* children is required to fully understand their perspective. This would shift the shared standards, beliefs, and attitudes indicated by societal norms. Play within digital spaces is examined from a range of different academic fields and thus different perspectives, however, little is known regarding how this research is approached when it comes to children's everyday lives, personal and ecological relevance, and the methods used. This is important as the extent to which children's perspectives of play is determined by how the research is designed and approached. Therefore, when studying children's experiences of play, it is important that the research is centred around the context of children engaging in play that they would typically engage in with devices and spaces that are available and accessible to them. This indicates the validity of play that is embedded within the temporal and spatial dimensions of daily life or the lived experience of the child [10]. This era of digitalisation is proffering children a range of options and choices regarding their play, therefore understanding the personal relevance of play in digital spaces as perceived by the individual indicates a clear purpose and meaning to those involved. Research with children should also denote an ecological relevance. An ecological relevance recognises children engaging in play as they typically would within their everyday digital spaces opposed to simulated research contexts [11]. The methods used throughout research should transition from conceptualising children from being passive participants to active consumers of digital devices and online worlds [3]. A plethora of creative and engaging data collection tools ensures children are listened to thus supporting successful participation and

acknowledging them as meaning-makers [4,20]. This acknowledges children as agentic agents who are able to make sense from their participation in play [4]. Similarly, the use of direct quotes from the child indicates that the child has been listened to and their opinion valued [20–22].

By studying how play in digital spaces is approached, gaps are identified to highlight areas for future development of research and knowledge in the area. This scoping review aims to describe and review the qualitative research articles which explores *children's experiences* of play in digital spaces from infancy through to adolescence. More specifically, how children's play in digital spaces is *approached* in research concerning, (a) the general relevance to children's daily life, (b) the personal and ecological relevance, and (c) the methods used.

## Methods

A scoping review was completed to expose and map the extent of the evidence available in the emerging and diverse field of children's experiences of play in digital spaces [23]. The review drew on the Preferred Reporting Items for Systematic reviews and Meta-Analyses extension for Scoping Reviews (PRISMA-ScR) and Arksey & O'Malley's (2005) methodological framework to provide a structured and systematic approach to the review [24]. Further recommendations were provided by the Joanna Briggs Institute [25]. The scoping review was constructed of 5 stages [26] which will be further described.

### Stage 1: Identifying the research question(s)

The scoping review will examine what evidence is available regarding children's perspectives of play in digital spaces by exploring how the daily relevance, personal and ecological relevance, and methods are approached within the research articles.

### Stage 2: Identifying relevant studies

To gather the most relevant literature, the first author consulted expert Librarians from University College Cork, Ireland and Luleå University of Technology, Sweden to review the search strategy. The primary search terms of *play*, *playing*, *game* or *gaming* were selected as these are what are typically used within the existing literature. These were combined using the proximity operator NEAR to locate hits that united play or gaming with digital or technology terms to ensure responses within a certain distance of each other regardless of the order. This initial search blocks were then combined using the Boolean AND operator to a second search block focusing on the concept of children's perspectives. The search blocks required thorough testing to ensure appropriate hits were returned due to the variety of terms used in literature from a broad range of academic fields. To ensure a thorough approach, a systematic search of four databases (Web of Science, ERIC, PsycINFO and Scopus) was completed to gather literature from a range of research fields, including Social Sciences, Education, Psychology as well as Computer Sciences. The electronic searches of databases were completed in October 2021 –see Table 1 for the final list of search terms. Research studies were restricted by publication in English (for pragmatic reasons) and limited to results within the last 15 years.

**Table 1. Search blocks.**

| Play OR playing OR games OR gaming | NEAR/ 2 | Digital OR technology OR virtual OR internet OR computers OR video OR online OR mobile OR applications OR "videogames" | AND | Child OR adolescent OR "young people" OR "young person" or teenager OR children | NEAR/ 2 | perspectives OR views OR opinions OR beliefs OR experiences OR perceptions OR understandings OR qualitative |
|---|---|---|---|---|---|---|

## Stage 3: Search selection

All references and abstracts were uploaded to Rayyan (www.rayyan.ai) from the four databases for the screening and selection process. This allowed for duplicate articles to be removed, resulting in a yield of 553 articles. Blind screening of first the titles followed by the abstracts was completed based on the inclusion and exclusion criteria by the authors (FL, BB). Inclusion criteria included qualitative papers between 2007–2021 and written in English for pragmatic purposes. Both authors met to discuss and agree on any divergent opinions in relation to the selection of the articles. A total of 63 articles were full text screened by all authors (FL, MLL, BB). Any studies not meeting the criteria were excluded and the remaining full-text studies were retained for data extraction. S1 Fig outlines the screening process of identifying and selecting of the study is indicated based on the PRISMA-ScR.

Following the initial screening and full-text review, a total of 32 articles were excluded based on the inclusion and exclusion criteria, specifically: research using quantitative data collection methods and/or analysis (n = 8), use of digital tools during real world play (n = 8), not peer-reviewed articles or original data collection (n = 6), child's perspective not clear (n = 3), focused on the design or evaluation of a specific game (n = 2), and not focused on play in digital spaces (n = 2). The articles excluded were done so following discussion by at least two of the authors. This resulted in a total of 31 articles retained for data extraction which met the inclusion and exclusion criteria.

## Stage 4: Charting the data

Two data extraction charts were created and reviewed iteratively throughout the extraction process to ensure all relevant information was presented. The first extraction chart focused on the content and the demographics of the research articles; the aim, the number and age of participants, and the data collection and analysis methods. The second data extraction chart was inspired on theories in relation to play and ecology [10,11,26,27] to facilitate a clear perspective on free play within an authentic and everyday context. The literature was systemically charted to scope the articles in relation to relevance to daily life, personal and ecological relevance, and the methods used. The **relevance to a child's daily life** signifies whether the games or play that was being researched is generally available in society to children and their families (on the open market allowing it to be downloaded and played on their own devices). Personal and **ecological relevance** refers to whether the games or play researched has purpose and meaning for the particular child's daily life and if they constitute a natural context in which children would engage in play experience. The **methods** applied for data collection and analysis ensures children's active engagement in the research, by the choice of data collection methods and how the results were presented such as their voice being clearly articulated.

## Stage 5: Collating, summarising, and reporting the results

The two data extraction charts were used by the first author to collate, summarise, and report on the results found. Firstly, the initial data extraction chart was used to collate the aim, the number and age of participants, and the data collection and analysis methods of the included articles. Following this, the second data extraction chart allowed for data to be drawn from the literature that responded to the research question. The data was extracted by thoroughly examining and analysing the content of the included articles. Categories then evolved iteratively by comparing and analysing the content of the articles in parallel to the data extraction charts to find commonalities and similarities between them [24]. Every step taken was discussed and reviewed between the authors (FL, MLL). Finally, all authors scrutinised, discussed, and validated the review to ensure that the results were grounded in the articles.

The analysis of content revealed that the research approached play in digital spaces in a variety of ways. The terminology of videogaming was most commonly used to describe the action of children's engagement with the activity. Few articles used the terminology of play, with none conceptualising play for play's sake as a right [1]. Thus, within the included articles play in digital spaces is most commonly referred to video games and will be consequently used throughout this scoping review.

## Results

### Evidence characteristics

Fifteen of the 31 articles were published in the last 5 years (2017–2021) and the majority of the studies originated from The United States of America (n = 11, 35%) and Europe (n = 11, 35%) (Table 2). The age of the children ranged from 3 to 18 years of age with studies (n = 18) tending to include older children (age 10 and above). Thirteen studies included children under the age of 9 years of age and of them, only three studies included children under the age of 6 years. Seven studies included an age range of children spanning 5 years or more. No studies included children with a recognised or diagnosed disability.

The analysis of the content of the articles revealed that the articles formed four categories (Table 3) of how they approached play in digital spaces: 'Videogames, behaviours, and societal norms', 'Videogames for its own sake', 'Videogames for learning', and 'Active Videogames for health promotion' that will be presented below. The results of the analysis within each category will be described by the general relevance to a child's daily life, the personal and ecological relevance, and the methods used.

### Video games, behaviours, and societal norms

This category includes a total of fourteen studies with five sub-categories focusing on various behaviours and societal norms which include articles on children's perspectives of video games influencing: social behaviours, cheating, violence, addiction, flow, and gender.

**Social behaviours in video games.** This category includes four articles [18,49,54,56] that explore children's experience and perspectives of the social, metacognition and self-scaffolding behaviours in video games. All studies demonstrated general relevance to a child's daily life by using video games that are popular and readily on the market, such as Fortnite, Clash of Clans, and Minecraft. The personal and ecological relevance was demonstrated for three studies. Participants were recruited who engaged regularly in video game play and the articles examined the participant's everyday play experiences indicating purpose and meaning for the child. The participants chose and engaged in their everyday play experiences demonstrating ecological relevance [18,49,56]. In Van Rooij et al's [54] study families were provided a console system, motion sensors and example games for a two-week period and were interviewed before and after [54]. By this, the purpose and meaning of videogaming was derived from the research question, focusing on the persistence of engagement in video game play which limited the personal and ecological relevance. All studies utilised interviews in data collection with the inclusion of observation [49] and observation and paper-based activities [56]. All articles in this category demonstrated the use of quotes throughout the articles to clearly position the child's voice throughout. Although one study presented the parents perspective before the child, extensive direct quotes from both children and parents were used throughout the results [18].

**Cheating and metagaming in video games.** Three studies explored children's perspectives of cheating or using metagaming in video games [39,44,50]. Metagaming indicates the pregame strategising which ultimately aims to advance game play by developing skills and techniques [58]. The general relevance to children's daily life was clear in all studies with

**Table 2. Characteristics of included studies.**

| Reference | Aim | Participants | Country | Data Collection | Data Analysis |
|---|---|---|---|---|---|
| Albarello, F., Novoa, A., Sánchez, M. C., Velasco, A., Hueyo, M. V. N., Narbais, F. [18] | To explore the social dimension of Fortnite and how it impacts children and their parents' perceptions of its use | 82 children (9–18 years old) from 32 households | Chile & Argentina | In-depth Interviews | Thematic analysis |
| Balmford, W. & Davies, H. [28] | How Minecraft on mobile devices is played and perceived within homes | 8 children (6–14 years old) from 5 households | Australia | Ethnography: informal interviews, play sessions, and participant observation | content |
| Barreto, D., Vasconcelos, L. & Orey, M. [29] | To explore student motivation and engagement levels in playing math video games | 2 children (8 & 9 years old) | USA | Screen recordings (primary data source) & closed and open-ended interview | Interaction analysis and Grounded Theory methods |
| Bassiouni, D.H. & Hackley, C. [30] | To investigate childrens experience as consumers of video games and associated digital communication technology | 22 children (6–12 years old) | UK | Focus groups and in-depth interviews | Discourse analysis |
| Brownell, C.J. [31] | To examine how the boundaries of the digital were blurred in response to a standardised writing prompt | 1 child (10 years old) | USA | Observation, field notes, writing samples, photographs, and lesson plans. Included formal interviews. | Iterative process |
| Carter, M., Moore, K., Mavoa, J., Gaspard, L. & Horst, H.[32] | To examine how children encounter and attempt to negotiate game addiction discourse | 24 children (9–14 years old) | Australia | Semi-structured interviews | Constructivist grounded theory techniques |
| Carter, M., Moore, K., Mavoa, J., Horst, H., & Gaspard, L. [33] | To explore what Fortnite offers young people as they move from children's gaming into genres that appeal to tweens and teens | 24 children (9–14 years old) | Australia | Semi-structured interviews | Not described |
| Daneels, R., Vandebosch, H. & Walrave, M [34] | To examine the ability of digital games to elicit meaningful or eudemonic experiences among adolescents | 33 children (12–18 years old) | Belgium | Focus groups and semi-structured interviews. | Horizontal analysis, Inductively and deductively |
| De Vet, E., Simons, M. & Wesselman, M. [35] | To explore children and parents opinions about active and non-active video games | 46 children (8–12 years old) 19 parents | Netherlands | Semi-structured focus groups. | Content analysis |
| Dixon, R., Maddison, R., Mhurchu, C. N., Jull, A., Meagher-Lundberg, P. & Widdowson, D. [36] | To explore children's and parents' perceptions of active video gaming | 37 children (10–14 years old) 27 parents | New Zealand | Focus groups. | Not described |
| Dodge, T., Barab, S., Stuckley, B., Warren, S., Heiselt, C. & Stein, R. [37] | To examine children's experiences in their participation in Quest Atlantis | 4 children (9–12 years old) | USA & Singapore | Participant observation and semi-structured interviews. | Constant-comparative analysis |
| Fonseca, R.M.G.S., Santos, D.L. A., Gessner, R., Fornari, L.F., Oliveira, R.N.G. & Schoenmaker, M.C. [38] | To identify and analyse the perception of high school students about sex, sexuality, and violence in intimacy relations, in light of the gender category | 27 adolescents (age not specified) | Brazil | Discursive commentaries from playing | Thematic content analysis |
| Hamlen, K. & Gage, H. [39] | To understand how students participate in and experience various methods of game play that don't follow traditional formats | 3 children (14, 15 & 17 years old) | USA | Phenomenological approach: exploratory study using in-depth semi-structured interviews | Thematic analysis |
| Hannaford, J. [40] | To explore children's imaginative interaction with Internet games | 8 children (8 & 9 years old) | A European City | Semi-structured interviews | Grounded theory approach |
| Huh, Y.J. [41] | To explore young children's digital game play outside the home | 4 children (3 years old) | USA | Observation and informal interviews, including field notes, photography, and videotaping | Bakhtinian interpretative analysis |

(*Continued*)

**Table 2.** (Continued)

| Reference | Aim | Participants | Country | Data Collection | Data Analysis |
|---|---|---|---|---|---|
| Inal, Y. & Cagiltay, K. [42] | To explore children's flow experiences in an interactive social game environment | 33 children (7–9 years old) | Turkey | Interviews and observation | Not described |
| Iqbal, A., Kankaanranta, M., & Neittaanmäki, P. [43] | To explore the experience and motivations of young people participating in virtual worlds | 15 students between (13–15 years) | Finland | Mixed methods: Questionnaires, Interviews, and observation | Not described |
| Kahila, J., Tedre, M., Kahila, S., Vartiainen, H., Valtonen, T. & Mäkitalo, K. [44] | To explore children's metagame activities | 142 children (12–15 years old) | Finland | Essay writing | Qualitative content analysis |
| Kutner, L.A., Olson, C.K., Warner, D.E. & Hertzog, S.M. [45] | To explore and identify themes in parents' and children's perspectives on video game play | 21 children (12–14 years old) 21 parents or guardians | USA | Focus groups using printed colour images to stimulate discussion | Thematic analysis |
| Leonhardt, M., & Overå, S. [46] | To quantify gaming and to examine how gender differences are perceived | 25 (13–16 years old) | Norway | Mixed methods: survey and semi-structured group interviews | Chi-square tests and thematic analysis |
| Maine, F. [47] | To explore what children's gaming orientations are as they play a digital narrative game. | 8 children (11 year olds) | UK | Observation and post-play discussion | Not described |
| Mertala, P. & Meriläinen, M. [48] | To explore what aspects of digital games appear meaningful for young children | 26 children (5–7 years old) | Finland | Drawing and informal interview | Descriptive analysis and interpretative analysis |
| Monem, R. [49] | To explore the metacognition and self-scaffolding processes involved in navigating digital immersive environments | 1 child (16 years old) | USA | Participants observation, face to face interviews, and document analysis of a cultural artifact | Content analysis |
| Nease, B., & Samura, M. [50] | To explore adolescent gamers perceptions about cheating | 12 children (14–17 years old) | USA | Semi-structured interviews | Inductive analysis |
| Olsen, C., Kutner, L., & Warner, D.E. [51] | To examine how children perceive the uses and influence of violent interactive games. | 42 boys (12–14 years old) | USA | Focus groups | Not specifically described |
| Sarachan, J. [52] | To explore how children's virtual worlds appeal to different players with varying levels of cognitive and social development | 16 children (6–11 years old) | USA | Observation and semi-structured interviews | Qualitative text analysis software |
| Soek, H.J., Lee, J.M., Park, C., & Park, J.Y. [53] | To explore adolescents' motivations for internet games. | 10 boys (12–17 years old) | South Korea | Photovoice and group discussion | Continuous comparison analysis |
| Van Rooij, A., Daneels, R., Liu, S., Anrijs, S. & Van Looy, J. [54] | To review three popular theoretical perspectives that cover motives for video gaming | 20 x family units covering data from 37 children and their parents (4–12 years) | Belgium | Interviews | Open, axial, and selective coding |
| Verenikina, I. & Kervin, L., Rivera, M.C., & Lidbetter, A. [55] | To explore how young children respond to the applications for mobile digital technologies offer varying opportunities for play | 10 children (3–5 years old) | Australia | Observation | Not described |
| Willett, R. [56] | To examine online gaming practices of children in home settings | 11 children aged 7–11 years | USA | Semi-structured interviews, paper-based activities, and observation | Thematic analysis |
| Willett, R. [57] | To examine families' everyday practices connected with online games played by children | 8 households. 5 x girls & 6 x boys (aged 7–11 years) who played online games | USA | Semi-structured interviews, paper-based activities, and observation | Thematic analysis |

**Table 3. Overview of data extraction.**

| | Daily Life | Personal & Ecological relevance | combination of methods | Methods | | | | Use of Quotes |
|---|---|---|---|---|---|---|---|---|
| | | | | Interview/ focus group | Observation | Drawing/ writing/ photographs | Other | |
| **(1)** Video games, behaviours, and societal norms (n = 14) | | | | | | | | |
| *Social behaviour (n = 4)* | 4 | 3/4 | 2/4 | 4 | 2 | 1 | | 4 |
| *Cheating (n = 3)* | 3 | 2/3 | 0/3 | 2 | | 1 | | 3 |
| *Violence (n = 3)* | 2 | 2/3 | 1/3 | 2 | | | 1 (discursive commentaries) | 3 |
| *Addiction (n = 2)* | 2 | 1/2 | 1/2 | 1 | | 1 | | 2 |
| *Flow (n = 1)* | 1 | 0/1 | 1/1 | 1 | 1 | | | 1 |
| *Gender (n = 1)* | 1 | 0/1 | 0/1 | 1 | | | | 1 |
| **(2)** Video gaming for its own sake (n = 10) | 10 | 5/10 | 6/10 | 9 | 6 | 3 | | 8 |
| **(3)** Video gaming for learning (n = 5) | 5 | 2/5 | 4/5 | 4 | 3 | 1 | 1 (screen recordings) | 5 |
| **(4)** Video games for health promotion (n = 2) | 2 | 1/2 | 0/2 | 2 | | | | 2 |
| **TOTAL** | 30/31 | 16/31 | 15/31 | 26/31 | 12/31 | 7/31 | | |

metagaming and cheating widely recognised as a dimension of videogame play. However, the personal and ecological relevance was not consistently demonstrated across all three. In two of the studies participants were selected due to their experience of playing video games [39,50]. It was not clear, however, whether all participants in Kahila et al., [44] study were regular players of video games thus whether it was of personal relevance. All three, however, did demonstrate ecological relevance by examining children's experiences of their metagaming and cheating in their real everyday video game play. All articles demonstrated data collection methods that actively engaged children in the research; Semi-structured interviews were utilised for two studies [39,50] and essay writing for the other [44]. The child's perspective was articulated throughout the results and their voice were reflected through the use of direct quotes.

**Violence in video games.** Three articles fall into the category of how children experience and perceive violence when playing video games [38,45,51]. Two studies [45,51] demonstrate general relevance to a child's daily life by focusing on video games that are available readily on the market, however, it was not clear from the article or from a search online whether the game in one study [38] was readily available and therefore relevant to a child's daily life. The personal and ecological relevance was valid for two studies [45,51] where participants were regular players of two or more of the bestselling violent games or had played video or computer games for at least 2 hours a week. In the remaining article [38] the game was only made available to the participants once they signed up for the research and no information was given if the game demonstrated personal relevance to the participating children's everyday life. However, this article did examine the participants experiences as they engaged with the game and it was not clear whether the play was directed from the researcher or the participants. Two studies used focus groups to generate data [45,51] with the final study extracting the discursive commentaries from the participants [38]. Despite Kutner et al., [45] positioning parental perspectives before the children in the results, the child's perspective was evident and direct quotes were used to help counterbalance this.

**Addiction in video games.** Two articles [32,53] examined children's perspectives of negotiating and understanding addiction when playing video games. Both studies demonstrated general relevance to children's daily lives as it concerned gaming on the internet [53] and the game, Fortnite, which is readily available on the market [32]. The personal and ecological relevance differed with one study recruiting regular players [53] and the other recruiting participants who were not all video game players [32]. The context for Seok et al [53] study was a treatment centre for adolescents receiving treatment for internet game addiction [53]. Both articles used methods to enable the children to talk freely about their own play experiences. Specifically, the use of Photovoice methodology provided participants the opportunities to decide the questions and topics for discussion within the category of addiction [53]. The use of semi-structured interviews [32] elicited the participants experiences of negotiating addiction discourses. The results from both studies clearly articulate the child's perspectives in the article and through the use of quotes.

**Flow in an interactive videogame.** One article examined children's flow experiences [42] of an interactive social videogame. This article demonstrated general relevance to daily life focusing on games that are readily available. However, the personal and ecological relevance was limited as not all participants selected were allowed or able to play computer games at home. Despite being able to choose and play games according to their own personal preference, the context in which it was examined was their computer lab at school. Qualitative and quantitative methods were used for data collection with a greater focus on the qualitative components. Despite the use of observation and interview, the children's perspective and voice were not clear throughout the results.

**Gender differences in videogaming.** One article [46] examined how students perceive and understand gender differences in videogaming which indicated general relevance for a child's daily life. The personal and ecological relevance varied as participants ranged from those with little gaming experience to those who referred to gaming as a passionate hobby. The use of focus groups facilitated the children to share their everyday experiences of videogaming and how they understand gender differences. The use of quotes positioned the child's voice and perspective throughout the results.

## Videogaming for its own sake

A total of ten articles fall within the autotelic play category [28,30,33,34,41,43,48,52,56,57]. All articles within this category demonstrate a general relevance to a child's daily life with games, devices, and apps used in the research readily available, such as Minecraft, Fortnite, and Club Penguin. The personal and ecological relevance were missing or unclear in five studies. Of those, not all the participants were regular video game or virtual world players in two studies [33,52]. A further two studies did not clearly articulate whether playing video games were relevant to the participants involved [28,48]. In the remaining article, despite the children being regular players at home, they were provided with individual iPads with purposively selected applications for play by the research team when they attended a Digital Playgroup [55]. This indicates a simulated context which resulted in a diminished personal and ecological relevance. The other remaining five studies demonstrated personal and ecological relevance as all children engaged in these activities in digital spaces at home.

A variety of data collection methods were used including, interviews, focus groups, drawing, and observation. Six articles used a combination of methods, resulting in four articles choosing one data collection method. Out of the ten articles, the methods used for three articles suggest that the child participants were not actively engaged in the research process with the researcher conducting research *of* children and dominating the voice of the child

[28,52,55]. For example, one article relied solely on the observation of children [55] resulting in the adult's perspective being elevated above the children. The remaining seven articles clearly positioned the child's voice by using direct quotes throughout the article.

## Video games for learning

Five articles explored children's experiences of the use of video games as a means to enhance children's learning and literacy [29,31,37,40,47]. The video games used; Minecraft, Club Penguin, and Quest Atlantis all have general relevance to children's daily life. Two articles demonstrated personal relevance with participants recruited as they were regular players of the game [31,40]. The further three articles did not clearly state whether the participants were regular players of the game [29,37,47]. Ecological relevance was established for three articles as children engaged in or discussed their typical play experiences [29,37,40], however, the remaining two articles were conducted in simulated contexts lacking ecological relevance [31,47]. Despite the participants in Maine's [47] study being able to choose who they played with, this involved playing a single-player game on a single device in pairs indicating a simulated context for the purpose of the research. Brownell [31] examined the participants' writing about experiences of video game play opposed to the playing. Observation and interviews were used in all studies [29,37,40,47], with the additional of written text in Brownell, [31] and drawings in Hannaford [40] study. The child's perspective and voice was clear throughout the results in four articles [39,44,56,57].

## Active video games for health promotion

This category includes two articles [35,36] that explore children's perceptions of active video games as a means for health promotion by addressing inactivity and obesity. The video games focused on both articles have general relevance to children's daily life but the personal and ecological relevance for the included children differed. The active video games were ecologically relevant for the children in [35] study as they were regular players, which indicated meaning and purpose for the children participating in the research. Although eligibility for inclusion required the participants in Dixon et al's [36] study to be either current or previous players of electronic console games, it was not specified whether they had previously played *active* video games. Therefore, the children were given a demonstration of active video games and the opportunity to practice playing the games up to a half hour before data collection. Both articles utilised focus groups for children to share their experiences of how they choose to engage in active video gaming, independent if they had played active video games before or if they just faced it before the data collection. The children's perspective was clearly described in the result and through the use of quotes in both articles.

## Discussion

In this scoping review we set out to expose and map the extent of the evidence available in the emerging and diverse field of children's experiences of their play in digital spaces. Specifically, we sought to examine the evidence of representations of digital play that has personal and ecological relevance through being embedded in the daily lives of children and families. Of the total thirty-one research articles identified we found that a total of twenty-one focused on play in digital spaces as a means for studying secondary constructs such as fostering specific behaviours, societal norms, learning, or for health promotion. Those articles addressing play for learning, for example, examined how video games engage children in their literacy practices in school [31]. Within the wider literature, it is acknowledged that video games incorporate a range of principles required for learning and thus, enhance learning [59–61].

Despite abounding concerns linking the use of videogames to inactivity and negative health consequences for children [6,16], the review identified a number of studies that sought to leverage the play dimension of active videogames to encourage a reduction in sedentary behaviours and obesity [35,36]. Similarly, other studies focussed on a genre linking play, videogames and health, namely, exergames and increasingly Virtual Reality games, can be seen to demonstrate how the active, play potential of gaming technology can reap health dividends for children, mirroring research in outdoor play [62,63]. As is the case with research focused on play in other spaces such as playgrounds [64], the findings from this review suggest that play as a dimension of videogaming can serve as a valuable function in supporting children's skills and development more generally [18,31,46].

The remaining ten articles centred on children's play experiences that could be considered 'free' or devoid of secondary purposes; play for the sake of play. Closer examination however, revealed that although these ten articles focusing on free play experiences had a relevance to children's daily lives, it was not clear whether five of the studies recruited participants who were regular video game players or for whom play in digital spaces might be an existing activity in their daily repertoire. This suggests that although the studies themselves examined play experiences that were free from the assumption of secondary effects or benefits, they did not present the same meaning or personal relevance to the children. The play experiences reported across these ten studies were further examined in terms of whether they were conducted in either a simulated or natural context. Of the ten articles in this study that focussed on 'free' play, seven were situated in environments that had ecological relevance, reflecting typical lives of young children by conducting research in contexts where they would typically engage in this type of play. Simulated research contexts included a specific Digital Playgroup [55], a college conference room [52], and a school computer lab [43] indicated a lack of ecological relevance.

The importance of examining play as embedded in space and time as well as ecologically situated in the lived experience of children recognises the complex transactional nature of the activity [10,11,65]. In research focussed on outdoor play or play in the home, the need to examine the experience of play where it occurs is strongly emphasised [66]. The findings from this review also suggest that any examination of play in digital spaces also requires that research be situated in authentic, ecologically valid contexts that are reflective of the personal, lived experiences of children. Despite this recognition of the benefits of researching digital play in naturally occurring contexts, careful attention must be given to understanding how best to capture play experiences from the perspective of the central protagonist, namely, the child. The studies that comprised this review tended to limit children's engagement in the research to participation modes more typically employed in research with adults such as focus groups [67,68] and interviews [69].

Recent work has highlighted an increasing imperative to examine the experiences of children from their own perspectives [70], in particular, those experiences that quintessentially belong to children such as play [71]. Despite a broad acknowledgments that research with children requires a shift from traditional methods of discursive data gathering to creative, age appropriate and child-directed techniques [4] only four of the studies in this review employed research approaches that might be considered 'child-centred' [28,41,48,57]. These results indicate that a more complete exploration of children's perspectives of play in digital spaces must focus on participatory, child-centred study designs that captures their unique experiences of free play that is embedded within their everyday lives. For example, the mosaic approach enables researchers to use a plethora of creative visual tools to ensure children are listened to thus supporting successful participation and acknowledging them as meaning-makers [4,20]. Techniques such as Photovoice or the use of drawings can also provide children with a visual

representation and enables researchers to elicit and generate children's narrative of participation in play thus providing participants with autonomy [72,73]. Furthermore, the scoping review highlights that to date, children from more diverse backgrounds or with particular needs, for example, those with disabilities have been largely ignored. Our findings also demonstrate that the majority of the literature focuses on older children [26,44] and that those younger children rarely received the primary focus during recruitment [41]. Considering the developmental nature of play in childhood [74] a greater focus of studies involving younger children and children from diverse populations' experience will provide greater depth of insights as to the nature of free play in digital spaces. Despite the limited availability of literature focussed on play for play's sake in digital spaces, a number of characteristics were evident that highlights the ways in which the experience of digital play resembles more traditional forms and some ways in which they diverge.

As noted earlier, play is widely accepted as a child's primary activity and is characterised as being initiated, organised, and controlled by the child themselves [8]. Within the literature, there are instances outlining that children are initiating and controlling the play themselves, such as the researcher observing the child's typical, everyday experiences within digital spaces [41,52] or through the consideration of the use of YouTube or Discord which allows children to plan strategies and techniques for gameplay [18]. Across some of the studies in this sample, it was reported that children enjoyed being able to play familiar aspects of games represented, but also being able to challenge themselves throughout the process of the game [33,49,52,54], thus providing a feeling of achievement which served to keep them engaged and interested. The ability to invent or to create new games such as playing hide and seek within the virtual space indicate that children respond to the imaginative dimensions within the virtual space. This indicates that when children are provided with the opportunities, they are able to go beyond the pre-defined rules and purpose of the game [47,48]. This mirrors the literature in nature and outdoor play that suggests that free play enhances their imagination and that they respond to the sense of achievement within digital spaces [75].

Reflecting on some of the emerging play opportunities brought about by recent developments in online gaming, virtual worlds and augmented reality, some studies suggest that children increasingly view digital spaces as social contexts in which they can maintain existing as well as create new friendships [18,33,44]. Although parents and children are reported as having divergent views on the role of play in digital spaces; as more time is spent online since the COVID-19 restrictions [76], online games, in particular, now afford considerable elements of social interaction within their design [77]. For example, 'Playground' mode in the popular building game, Minecraft, presents opportunities' to 'hang out' augmenting the extension of children's offline social environment [33,46]. This reflects some findings from research that explores children's play in other domains such as outdoor settings or risky play that play can enhance social skills [64,75]. Despite the limited studies available in this review, it is interesting to note that children fully understand the potential such game experiences offer them when playing with other children either when co-located with each other or, increasingly when remotely located [77,78]. For example, many games, such Fortnite and Among Us, that support cross-platform play promote opportunities for children to engage in social play without the limitations of traditional, offline play spaces such as homes, schoolyards and playgrounds. Thus, recognising the considerable social dimensions of games within digital spaces proffer a complex and interactive play experiences for children that is comparable to the experiences offered within offline contexts.

Although representations of play for children that are freely chosen are evident across the studies in this review, these experiences presented are not exclusively described as play. Rather

there is an interchangeability of language used to describe what it is that children do when engaging in such activities.

The studies included in this scoping review referred to experiences that may be construed or interpreted as children's play. It is evident, however, that the continued use of descriptions such as 'digital games' [34,48], 'video gameplay' [50] and 'videogames' [30,51] obscures our understanding of what truly constitutes play as an autotelic, freely chosen children's activity in the context of digital spaces. Furthermore, it highlights the need for researchers to focus more on the individual perspectives of children and to trust their ability to represent their own experiences using a language that is familiar to them and reflects their own individual interpretation of what they see as 'play' in digital spaces.

## Limitations

A systematic and rigorous approach was adopted [24,79] when carrying out this scoping review. However, some limitations exist. The criteria of excluding all but peer-reviewed articles as the primary source of data may have missed worthy perspectives from alternatives such as conference papers. The review included papers in English language only, therefore evidence published in other languages may have been missed. A total of four electronic databases were selected and searched, and despite those covering a range of academic fields, articles from other databases may have unintentionally been excluded. Finally, due to the diversity and scope of the field, the use of search blocks and specific terms, may have resulted in exclusion of further research articles. To counteract this, various test searches were completed with advice from experts in the field and Librarians. Initial articles found to be of interest were examined for keywords in their title and abstract to ensure the use of appropriate terminology. However, it is important to acknowledge that despite these actions relevant articles may have been excluded, therefore, to compensate for this, a hand search of the literature was completed.

## Conclusion

This scoping review aimed to describe and review the qualitative research articles which explore children's experiences of play in digital spaces. Specifically, we sought to examine the evidence of representations of play in digital spaces that has personal and ecological relevance through being embedded in the daily lives of children and their families. Children's experiences of play in digital spaces has been utilised by researchers to explore a variety of categories, from children's learning, health promotion, to societal norms and behaviours. The focus of research articles on free play or play devoid of secondary purposes did not consistently demonstrate personal and ecological relevance to the children. To extend the knowledge regarding children's perspectives of play in digital spaces and to begin to shift societal discourses, research must be contextualised and clearly positioned from the perspective of child [4,11,26]. This will help alter adults' perspectives of the opportunities of digital spaces from the current discourse which is dominated by the negative consequences of screen time. In turn, elevate free play in digital spaces as an appropriate choice for children and their families.

This scoping review has identified a lack of articles focusing on children's experiences of play in a digital space, and these perspectives are essential for parents, professionals, game designers, and policymakers alike to develop an enhanced understanding of play within this emerging space. For appropriate guidelines to be suitably developed for health promotion, an in-depth, thorough, and clear understanding of children's experiences of play opposed to their use of screen time or any other passive engagements with technology. Play as an activity that is characterised by being initiated, organised, and controlled by the child themselves [8] is evident in both traditional settings of playgrounds and the outdoors but also within digital

settings. Despite this, these experiences within digital spaces were not exclusively described as play within the literature. Thus, a change of language is therefore essential to ensure that children's experiences of their primary activity is interpreted and described in language familiar to them.

By focusing our attentions of how research is approached, a child-centred approach can be adopted ensuring research is relevant to children's daily life, ecologically relevant, focused on the occupation, and utilising methods of data collection maximising the child's voice to be listened and shared.

## Supporting information

**S1 Fig. PRISMA flow diagram.**
(TIF)

## Author Contributions

**Conceptualization:** Fiona M. Loudoun, Bryan Boyle, Maria Larsson-Lund.

**Formal analysis:** Fiona M. Loudoun.

**Methodology:** Fiona M. Loudoun.

**Supervision:** Bryan Boyle, Maria Larsson-Lund.

**Writing – original draft:** Fiona M. Loudoun.

**Writing – review & editing:** Bryan Boyle, Maria Larsson-Lund.

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
