## [Decision Letter · Decision Letter 0]

19 Apr 2022

PONE-D-22-03218Children's experiences of play in digital spaces: a scoping reviewPLOS ONE

Dear Dr. Loudoun,

Thank you for submitting your manuscript to PLOS ONE. After careful consideration, we feel that it has merit but does not fully meet PLOS ONE’s publication criteria as it currently stands. Therefore, we invite you to submit a revised version of the manuscript that addresses the points raised during the review process.

We look forward to receiving your revised manuscript.

Kind regards,

Marc Potenza

Academic Editor

PLOS ONE

Journal Requirements:

Reviewers' comments:

Reviewer's Responses to Questions

**Comments to the Author**

1. Is the manuscript technically sound, and do the data support the conclusions?

Reviewer #1: Yes

Reviewer #2: Partly

2. Has the statistical analysis been performed appropriately and rigorously? 

Reviewer #1: N/A

Reviewer #2: N/A

3. Have the authors made all data underlying the findings in their manuscript fully available?

Reviewer #1: Yes

Reviewer #2: Yes

4. Is the manuscript presented in an intelligible fashion and written in standard English?

Reviewer #1: Yes

Reviewer #2: Yes

5. Review Comments to the Author

Reviewer #1: Review

The scoping review conducted by the authors is very useful given the developments in digital play, the increasing focus on children's rights and the relevance of knowing children's expectations and experiences in this regard and including them in further research. The article is logical and coherent, the aim clear and the methodology clear. I have a number of suggestions for the authors to consider:

1. The text refers to both play and free play - if both mean play in the sense of Article 31 UN CRC, then the addition of 'free' is not necessary (this is included in the concept of 'play' in the sense of voluntary and unscripted play); if, according to the authors, there is a difference between the two, then this could be explained.

2. I understand that a clear demarcation is needed, but I find it unfortunate that hybrid forms of play (combination of offline and online play) were not included, since the offline and online daily lives of children are now completely intertwined and they may no longer make a distinction in their play. In addition, I suspect that one would then also focus the perspective a bit more on the younger children as I expect that especially with them, the form of play is relevant (at least it would be interesting to find out). Perhaps this is something to elaborate on in future research.

3. The article contains a reference to a report by Livingstone and Pothong on playful by design; both scholars recently published an academic paper on the subject that the authors might want to mention in their article: https://www.tandfonline.com/doi/full/10.1080/1369118X.2022.2046128

4. I was interested in the 'shift in societal norms' that the authors refer to (see, among others, line 79) but they do not elaborate on it; perhaps this could be explained a bit more.

5. As part of the analysis, the 'personal and ecological relevance' (a.o. line 106) of digital play for children is examined. This is explained later in the article (see line 171), but I would bring the explanation forward. It was not clear to me what the authors meant by it so an earlier explanation would be helpful (and, as far as I am concerned, it could be explained a bit more).

6. From line 238 onwards, it talks about cheating and meta gaming: can the authors give examples of what is meant by that?

7. Finally, I would read the text again carefully because sometimes sentences did not seem to run quite right, see e.g.

Line 39

Lines 83-84

Lines 64-67

Line 257 (‘the’ missing at the start of this line)

Line 460

On a final note, It might also be interesting for further research to find out how play and digital play are different and if, from that point of view, research has been done into children's experiences, expectations and concerns. One difference, for example, is that digital play almost always takes place in a commercial environment (a factor that the Children's Rights Committee (and others) considers problematic for various reasons from the perspective of the right to play but also other children's rights). One example is the aforementioned study by Livingstone and Pothong, but there may be more.

Reviewer #2: This work focuses on a very important and topical issue since the use of digital devices by children and teens has become even more pervasive since the pandemic. I agree with the authors that the discourse largely remains from the perspective of adults and dominated by the negative consequences of increased screen time and sedentary behaviors. An article, therefore, focused on this topic is sorely needed and could contribute greatly to the scientific community. I suggest, however, that this article needs revisions to ensure that it contributes substantially to the discourse. The research question itself, especially the second question, is too vague and somewhat convoluted, "The scoping review will examine what evidence is available regarding children’s perspectives of play in digital spaces and how the daily relevance, personal and ecological relevance, and methods are approached within the research articles". Examing children's perspective of play in digital speaces is clear. Does the second question mean to examine methods used by the articles to examine the first question? If so, it shouldn't be "and methods". It may seem an irrelevant distinction but it makes the reading and purpose a little clearer. I would also like a clearer definition of "ecological relevance". It is used throughout the article and may mean different things in different scientific fields. The abstract notes that, "these perspectives are essential for parents, professionals, game designers, and policymakers alike" but there is little discussion of why this is essential. What will parents, for example, learn from their children's perspective of digital play that would impact their parenting? In general, the discussion piece would have benefitted from a clearer delination of the arguments (there was some repetition and unclear links between aspects of the analysis) and the relevance to the lives of children. The reader can get a little lost in the critique of the articles not included children's voices and wish for more analysis of how these digital platforms could/do provide opportunities for child centered play. There is some inclusion of this, I just wished for more clarity in some places and expansion in others.

Proof readings would be helpful i.e. line 484, 442, 512 etc.

6. PLOS authors have the option to publish the peer review history of their article (what does this mean?). If published, this will include your full peer review and any attached files.

Reviewer #1: **Yes: **Simone van der Hof

Reviewer #2: **Yes: **Dr. Margaret O'Donoghue

---

## [Author Response · Author response to Decision Letter 0]

17 May 2022

Dear Academic Editor and Reviewers,

Many thanks for taking the time to review the manuscript: PONE-D-22-03218, Children's experiences of play in digital spaces: a scoping review. Please see the responses below responding to each point of your feedback. The response is in italic directly below the comments made by each individual. We hope that the responses are to your satisfaction.

Academic Editor

1. Please ensure that your manuscript meets PLOS ONE's style requirements.

Manuscript has been cross-checked with the formatting guidelines and style requirements.

2. You have not specified where the minimal data set underlying the results described in your manuscript can be found.

This is not applicable for the scoping review – the minimal data set has been identified throughout the review. The databases and search terms have been fully described throughout the manuscript.

3. Should your manuscript be accepted for publication, we will hold it until you provide the relevant accession numbers or DOIs necessary to access your data.

As highlighted above, this will not be applicable.

Reviewer #1

1. The text refers to both play and free play - if both mean play in the sense of Article 31 UN CRC, then the addition of 'free' is not necessary (this is included in the concept of 'play' in the sense of voluntary and unscripted play); if, according to the authors, there is a difference between the two, then this could be explained.

Throughout the text, play is referred to as the play in the sense of the UNCRC, therefore, we agree that free is not required as an addition to play. Free has therefore been removed from the text where necessary.

2. I understand that a clear demarcation is needed, but I find it unfortunate that hybrid forms of play (combination of offline and online play) were not included, since the offline and online daily lives of children are now completely intertwined, and they may no longer make a distinction in their play. In addition, I suspect that one would then also focus the perspective a bit more on the younger children as I expect that especially with them, the form of play is relevant (at least it would be interesting to find out). Perhaps this is something to elaborate on in future research.

It is particularly evident from the research that children do not differentiate between online and offline play as adults tend to do. There is some research by Kate Cowan and John Potter (https://doi.org/10.1177/2043610620941527), examining how the two intersect. I believe that this will evolve organically as our research progresses.

3. The article contains a reference to a report by Livingstone and Pothong on playful by design; both scholars recently published an academic paper on the subject that the authors might want to mention in their article: https://www.tandfonline.com/doi/full/10.1080/1369118X.2022.2046128

Thank you for highlighting the academic paper by Livingstone and Pothong. This has been updated accordingly in the manuscript.

4. I was interested in the 'shift in societal norms' that the authors refer to (see, among others, line 79) but they do not elaborate on it; perhaps this could be explained a bit more.

This has now been elaborated on to discuss the binaries that play is often positioned within as a result of an adult dominated discussion.

5. As part of the analysis, the 'personal and ecological relevance' (a.o. line 106) of digital play for children is examined. This is explained later in the article (see line 171), but I would bring the explanation forward. It was not clear to me what the authors meant by it so an earlier explanation would be helpful (and, as far as I am concerned, it could be explained a bit more).

We have been aware of ensuring that terms are clearly defined given that they may mean different things to different professions. Lines 48; 85-92 define and explain personal and ecological relevance, however, the sentence has been altered to begin: ‘Personal and ecological relevance refers to…’ to ensure it is explicit.

6. From line 238 onwards, it talks about cheating and meta gaming: can the authors give examples of what is meant by that?

A definition of metagaming has been included as well as examples of cheating within a virtual world.

7. Finally, I would read the text again carefully because sometimes sentences did not seem to run quite right.

A further proofread of the manuscript has been completed.

Reviewer #2

1. The research question itself, especially the second question, is too vague and somewhat convoluted, "The scoping review will examine what evidence is available regarding children’s perspectives of play in digital spaces and how the daily relevance, personal and ecological relevance, and methods are approached within the research articles". Examining children's perspective of play in digital spaces is clear. Does the second question mean to examine methods used by the articles to examine the first question? If so, it shouldn't be "and methods". It may seem an irrelevant distinction, but it makes the reading and purpose a little clearer.

Thank you for the comment. The order of the words in the second part of the research question provides sequence and cohesion for the findings that follow. However, by adjusting the question to:

 “The scoping review will examine what evidence is available regarding children’s perspectives of play in digital spaces by exploring how the daily relevance, personal, and ecological relevance and methods are approached within research articles.” May help to make the distinction clearer.

2. I would also like a clearer definition of "ecological relevance".

This was also highlighted by Reviewer #1. From lines 85-92 – personal and ecological relevance is defined and described. However, the sentence has been reworded to ensure explicit defining of the terms. 

3. The abstract notes that, "these perspectives are essential for parents, professionals, game designers, and policymakers alike" but there is little discussion of why this is essential. What will parents, for example, learn from their children's perspective of digital play that would impact their parenting?

Sentence altered to consider that this will inform and contribute to an enhanced understanding of play in digital spaces.

4. In general, the discussion piece would have benefitted from a clearer delination of the arguments (there was some repetition and unclear links between aspects of the analysis) and the relevance to the lives of children. The reader can get a little lost in the critique of the articles not included children's voices and wish for more analysis of how these digital platforms could/do provide opportunities for child centered play.

A further proofreading of the document has been completed to ensure a clearer and more succinct analysis.

5. Proof readings would be helpful i.e. line 484, 442, 512 etc.

Proof reading of the full document has been completed.

---

## [Decision Letter · Decision Letter 1]

25 Jul 2022

Children's experiences of play in digital spaces: a scoping review

PONE-D-22-03218R1

Dear Dr. Loudoun,

We’re pleased to inform you that your manuscript has been judged scientifically suitable for publication and will be formally accepted for publication once it meets all outstanding technical requirements.

Kind regards,

Marc Potenza

Academic Editor

PLOS ONE

Additional Editor Comments (optional):

Reviewers' comments:

Reviewer's Responses to Questions

**Comments to the Author**

1. If the authors have adequately addressed your comments raised in a previous round of review and you feel that this manuscript is now acceptable for publication, you may indicate that here to bypass the “Comments to the Author” section, enter your conflict of interest statement in the “Confidential to Editor” section, and submit your "Accept" recommendation.

Reviewer #2: All comments have been addressed

2. Is the manuscript technically sound, and do the data support the conclusions?

Reviewer #2: Yes

3. Has the statistical analysis been performed appropriately and rigorously? 

Reviewer #2: Yes

4. Have the authors made all data underlying the findings in their manuscript fully available?

Reviewer #2: Yes

5. Is the manuscript presented in an intelligible fashion and written in standard English?

Reviewer #2: Yes

6. Review Comments to the Author

Reviewer #2: The authors have addressed all of the previous suggested revisions. The manuscript is, therefore, appropriate for publication. One minor editing requirement on Line 51, "Thiss" needs to be revised to "This".

7. PLOS authors have the option to publish the peer review history of their article (what does this mean?). If published, this will include your full peer review and any attached files.

Reviewer #2: **Yes: **Dr Margaret O'Donoghue

---

## [Editor Report · Acceptance letter]

1 Aug 2022

PONE-D-22-03218R1 

Children's experiences of play in digital spaces: a scoping review 

Dear Dr. Loudoun:

I'm pleased to inform you that your manuscript has been deemed suitable for publication in PLOS ONE. Congratulations! Your manuscript is now with our production department. 

Kind regards, 

on behalf of

Dr. Marc N. Potenza 

Academic Editor

PLOS ONE